# Factors Affecting the Use of Balanced Scorecard in Measuring Company Performance

**Eva Benková** [1],*, **Peter Gallo** [2] , **Beáta Balogová** [3] and **Jozef Nemec** [4]

1   Department of Intercultural Communication, Faculty of Management, University of Prešov, 080 01 Prešov, Slovakia
2   Department of Management, Faculty of Management, University of Prešov, 080 01 Prešov, Slovakia; peter.gallo.1@unipo.sk
3   Department of Educology and Social Work, Faculty of Arts, University of Prešov, 080 01 Prešov, Slovakia; beata.balogova@unipo.sk
4   Department of Economics and Economy, Faculty of Management, University of Prešov, 080 01 Prešov, Slovakia; jozef.nemec@unipo.sk
*   Correspondence: eva.benkova@unipo.sk

**Abstract:** The paper presents the results from the research on the factors influencing the use of the Balanced Scorecard methodology in measuring company performance in the engineering sector. The primary objective of the research was to verify the importance of using non-financial factors in managing businesses in connection to the use of the Balanced Scorecard methodology and to verify the dependence between the use of the given methodology and the lack of human and financial resources for its usage. The research focusing on the given issue was conducted over a period of six months. The research was based on the hypotheses that were verified with statistical methods using the methodology of a Chi-square test. To identify the factors that hinder the usage of the Balanced Scorecard methodology in the addressed enterprises, the method of standard deviation was used. The main result of the research is a finding that there is a statistically significant relationship between the enterprises considering the non-financial indicators and the use of the Balanced Scorecard methodology to be important. This relationship is confirmed also by the calculation using a test with $p = 0.0422$. The research verified one of the main research goals, i.e., the importance of non-financial indicators in connection to the Balanced Scorecard concept (BSC). Other hypotheses are related to the issue of the lack of human and financial resources. Using the Chi-square test in these cases once again, the study also found out the existence of the dependence between the lack of these resources and the use of the Balanced Scorecard methodology. The final value $p = 0.0446$ relating to human resources and the value $p = 0.0377$ relating to financial sources define the barriers as being important in implementing the BSC methodology into corporate practice. These values confirm other research results related to the barriers of using BSC. The presented paper assesses the research results that confirm the importance of using non-financial indicators and define the barriers that hinder this usage. The research contributed to the extension of the knowledge of the BSC concept that we consider being a modern managerial future-oriented tool and supported its implementation in companies so that they could operate within the framework of sustainable development.

**Keywords:** non-financial indicators; Balanced Scorecard; engineering; performance; sustainability

## 1. Introduction

In the current environment, there is a growing tendency for sustainable development of corporate activities through implementing innovations [1,2]. The turbulent and constantly changing conditions

in the production environment also entail certain risks. Traditional methods are no longer enough to respond to the diversity of client requirements and short product life cycles [3–5]. Strategic thinking is essential for achieving market success and competitiveness. Strategic thinking will enable enterprises to gain benefits in risk management, cost savings, efficient human resources management, and in particular to implement innovation policy [6–8]. The current period of development in the area of strategic management is based on managers' abilities to anticipate what will happen in the future and how a market will develop [9–11].

To improve company's performance, it is essential to monitor and respond correctly to current trends that affect its business. Only then, the company can make a profit and meet the expectations of the owners and other interest groups. Using strategic approaches will enable the company to better adjust the activities in its portfolio [12,13]. At present, ownership can be understood in two perspectives, both real ownership and managerial ownership [14].

The enterprise performance is determined by rising financial entitlements which are (to some extent) the result of globalization, interdependence and internationalization [15]. Both types of owners belong to the concept of the parties involved, while the direction towards the business environment has developed an approach that consists of the theory of the parties involved [16]. In this connection, nowadays, measuring the performance of enterprises comes to the fore. The issue of the method of monitoring and managing performance is a subject of discussion among experts who seek a solution how to create a managerial tool supporting constant improvement and evaluation of the performance of corporate processes.

Currently, measuring enterprise performance is characterized by two basic points of view on enterprise functioning. The first perspective focuses on the financial part that represents a financial investment for an owner when owners expect the appreciation and increase of the enterprise value. The second perspective is an enterprise understood as a socio-economic system, i.e., a complex net of internal and external relationships that need to be evenly directed. The Balanced Scorecard methodology is based on the second perspective and its main aim is to provide a strategic view on managing a business, given by a vision, mission and strategy including measuring the performance from financial as well as non-financial views.

Our research is oriented to non-financial indicators connected with the Balanced Scorecard methodology. The non-financial indicators are quantitative data expressed in other than monetary units or ratio indicators expressed in other than monetary units. These include indicators such as employees' satisfaction, productivity, safety and others [17]. Using the non-financial indicators of performance influences positively the performance of businesses in an uncertain environment. The companies using the systems of measuring performance (using subjective and objective non-financial indicators) achieve better performance and put greater emphasis on the quality of production (mainly those using subjective non-financial indicators). The non-financial indicators effects positively managerial performance with clearly defined tasks in an organization [18,19] states that enterprises focused on innovations use the non-financial indicators of performance more often.

This paper aims to point out the connection between financial and non-financial indicators that are part of the Balanced Scorecard methodology and to verify the factors related to the barriers hindering the use of this methodology. We assume that businesses are largely managed through financial indicators, but the current trend shows the increasing importance of non-financial indicators in conducting business activities. A number of complex systems have been created to measure the performance of the company that contain features of financial but also non-financial management. One of the systems is the Balanced Scorecard (BSC) concept which is the subject of our research. We conducted a research of the Balanced Scorecard methodology in the past but with a more detailed focus on using the indicators of the BSC methodology perspectives in small and medium-sized enterprises related to the importance of non-financial indicators [20]. We decided to expand the previous research with the barriers that hinder the use of BSC thus differentiating these two studies. The research was

also based on the limits established in the previous study which were restricted only to the segment of small and medium-sized enterprises.

There are only few studies on BSC methodology applied in the conditions of the Slovak Republic. With the given research, we would like to contribute to the expansion of the knowledge in this issue and to offer an insight in the environment of industrial businesses. It could be also seen as an opportunity to fill in the gap in the research focused on managerial tools in the conditions of Slovakia.

In this context, we have chosen the industrial sector, as this sector is one of the largest GDP- Gross domestic product producers in the country and in connection with the digitization of this industry it needs innovative impulses and the use of modern management tools. The BSC methodology has been the subject of investigation for some time, but in the context of the increasing need for innovations into non-financial indicators, we consider it suitable for improving business performance. The current trend, such as the digitization of industry, has motivated us to conduct research of Balanced Scorecard methodology in industrial enterprises.

The industrial sector is the driving force of the Slovak economy and contributes a significant percentage to GDP. From a global perspective, the most important challenges are challenges for the employees of companies who influence the implementation of the 4.0 Industry. These challenges include poor professional training and a lack of digital culture, a lack of clear vision, high financial requirements for investment, recruitment, and development of new talents, and a general reluctance of all parties involved to change [21–24]. Industrial enterprises seek to continuously improve in all areas of business and take advantage of opportunities, whether financial or non-financial in their favor [25,26]. The study by [27] identified the impact of industry 4.0 on business performance. This study has also prompted us to explore other variables that convey Industry 4.0's impact on business performance. As an industry-oriented country, Slovakia presents the potential for conducting research on management tools and comparing them with other developed countries. Based on this fact, the research on the Balanced Scorecard was especially focused on the industrial sector that we consider to be one of the pillars of a sustainable economy.

The presented paper is divided into individual parts. The first part of the paper deals with the introduction and the literature research focused primarily on the Balanced Scorecard methodology.

The second part is focused on the explanation of selected methods and description of data acquisition.

The third part deals with the research results and the verification of hypotheses that formed the basis of our research. The section contains statistical evaluations of hypotheses with graphical representation.

In the discussion part of the paper, the focus is paid to the already conducted research in the given issue and compares it with our results.

The end of the paper defines the paper contributions to the given issue and also set the limits of the research.

## 2. Literature Review

Customer-oriented businesses use non-financial indicators and balanced multi-criteria systems to measure their performance. The Balanced Scorecard concept is a comprehensive tool in which each organizational unit has to adapt its activities to achieve specific aims in relation to defining a business strategy [28,29]. Individual business units in companies need to identify their measurements to link the four BSC core perspectives. These perspectives include the customer perspective, the perspective of internal processes, the perspective of innovation and education, and the financial perspective [30]. Each perspective is defined by setting strategic goals in the given field. For the strategic objectives, measures are chosen which serve as a basis for quantitative control. It is also necessary to choose target values, and strategic actions, through which the company is to reach the set goals. Strategic goals, measures, target values and strategic actions are mutually interconnected by linkages operating on a cause-effect principle. The tasks defined in this way form the basic principle of the Balanced Scorecard concept. [31,32].

Balanced Scorecard helps visualize how the determined goals can be achieved and what success factors are necessary to reach the desired goals [33]. The BSC concept enables companies to get feedback from each organizational unit related to their control, which will help the business to achieve better financial performance and the ability to innovate in individual organizational areas [34]. The causal use of the chain increases the attention to the corresponding measurements and extends the search effort and the quality of decision making under the responsibility for the outcome. Contrary to that, we claim that the use of a causal chain within the process of responsibility reduces the search effort and does not cause a similar improvement in decision quality.

Balanced Scorecard is a concept of how to transfer vision and strategy into goals and their metrics so that they comprehensively cover not only areas of the company's financial performance, but also non-financial areas. The goals and measures are organized within the BSC into four perspectives—financial, customer, internal processes and learning, and growth. The BSC is a tool of communicating mission and strategy between the various levels of management and ordinary employees. It is used to keep all workers informed about the activators that affect current and future success [35,36].

The literature presents many problems in implementing the Balanced Scorecard concept. One of the main problems is that BSC is a tool that is too expensive for enterprises. This creates a significant hindrance for businesses wishing to apply this concept. The second important factor is the difficulty of its implementation. Each business has its own unique environment, and the implementation of BSC is not a universal issue that applies equally to all businesses. For the correct implementation of BSC, it is necessary to contact advisory and consulting companies that implement this concept for relatively high prices. It is therefore difficult to understand the most relevant criteria that are customizable to assess the total performance. The concept of BSC is also connected with the creation of a strategic map, which will enable better reading of strategic goals and ways of achieving them in the form of graphical visualization. Both the short-term and long-term objectives need to be taken into account when developing the strategic map. These objectives should then be communicated to all organizational levels of a company. There is also a problem with this procedure which has revealed that it is difficult to choose, correct and then communicate the aims that need to be accomplished. In fact, the absence of a connection between managers and objectives was identified when developing the Balanced Scorecard concept [37,38].

The main mission of the Balanced Scorecard concept is that the company is not managed according to the past but according to future-oriented strategies to secure its long-term existence. It is important to formulate a strategy that is comprehensive and that it covers all business areas so that it is specific and transparent, touches all specific workers and is motivating. When choosing the right goals and measures, Balanced Scorecard explains the strategic direction of a business while enabling its measurement. With an appropriate choice of objectives, Balanced Scorecard can guide the company's behavior complaint with the strategy since goals affect behavior. The main meaning of the Balanced Scorecard concept is that strategic goals and their representation are at the forefront. Strategic goals are derived from a vision and strategy and thus become strategically important goals of the company deciding on its overall success. To plan and monitor their achievement, it is necessary to assign the corresponding financial and non-financial indicators to these objectives and the target and actual values of these measures. The strategic actions that are assigned to each objective should ensure that the objectives are achieved. Therefore, each strategic action has a given deadline, budget, and a specific responsible person [39–41].

The main objective of the BSC concept is to rectify the constraints of traditional performance measurement tools, as well as to transform business strategies into key performance indicators (KPI) to ensure a balance between short-term performance measured through financial indicators and non-financial factors that should head the organization towards better competitiveness and long-term sustainability [42,43]. The model identifies optimum values that are used as crucial elements to attain

overall performance. This model then combines the intangible elements of performance measurement, which are vital for many organizations [44].

The methodology of the traditional BSC has evolved towards a business sustainability balance sheet which has integrated institutional, economic, socio-cultural, and environmental perspectives. In terms of sustainability, BSC determines a causal connection between business factors in order to set priorities and objectives in a rational process of making decisions [45].

Management tools including the Balanced Scorecard methodology are the subjects of the studies by experts and various consulting companies.

The issue of the BSC is regularly studied by [46] that conducts studies focused on its usage and the satisfaction in enterprises every year. As it results from their research in the year 2017, the Balanced Scorecard methodology is applied by approximately 38% of the addressed companies in the world. [47] presents in their research from 2018 that the Balanced Scorecard is used in more than 44% of medium and large enterprises. Another research specialized in the use of the Balanced Scorecard was carried out by [48] in 2018. It was aimed at the factors that hinder the use of the mentioned concept in addition to the use of the BSC. It identifies financial resources and a lack of qualified labor force as the main factors.

Foreign and Slovak authors use the questionnaire method as the main research method. This method is one of the most commonly used methods in research. It is used in social sciences to collect facts quickly and on a mass scale. In our research we needed to find out the opinions of respondents, companies, in an anonymous form in a certain period, and it is just this method that meets these prerequisites. BSC studies has been conducted using various statistical methods. The authors' research [49] was evaluated using the Spearman coefficient and methods of analysis, synthesis, and generalization. Studies conducted by foreign authors [50] also used statistical methods such as Kendallo tau, which is relatively identical to the Spearman R test, to examine management tools. In our research we used the method of proportion of the phenomenon in the population because we were not able to get answers from all respondents and we had to expect a certain deviation for the relevant output. The deviation in the mentioned method is determined by the degree of reliability. The calculation is given in the options from-to and the result can be interpreted using this range. We used Pearson's Chi-square test to verify the hypotheses aimed at determining dependencies. This degree of reliability of a relation between two categorical variables is most commonly used. The test is based on measuring the differences in the real frequencies in the contingency table cells compared to the expected ones, where the expected cell frequency is calculated as the proportion of the product of marginal frequency of the corresponding row and column and the total number of respondents. The significance of the Chi-square test increases as the measured differences increase. This test is one of the most common methods for testing hypotheses aimed at determining dependencies between variables used in our research, too.

Based on the given facts, the following hypotheses were defined in our research focused on non-financial indicators with regard to the Balanced Scorecard concept. These hypotheses are also characterized with the main reason for their determination. More detailed information can be found in the literature review and discussion.

**H1.** *There is a statistically significant relationship between the importance of non-financial indicators and the use of the Balanced Scorecard methodology.*

This hypothesis is based on the concept of the BSC methodology, which the authors [32] have based their research on, i.e., if a company wants to be successful, it must also use non-financial indicators to improve its performance.

In determining H2 and H3, we used research by companies [47,48,50] and others that focused on examining barriers to the introduction of management tools and also defined barriers related to BSC methodology.

**H2.** *There is a statistically significant relationship between the lack of human resources and the use of the Balanced Scorecard methodology in engineering enterprises.*

**H3.** *There is a statistically significant relationship between the lack of financial resources importance and the use of the Balanced Scorecard methodology in engineering enterprises.*

In determining,

**H4.** *The Balance Scorecard concept is used by more than 20% of industrial enterprises, we proceed from previous studies focused on using the BSC methodology in the domestic environment, in a near region and abroad.*

The consulting company [46] conducts annual research on management tools, including Balanced Scorecard methodology. In its research, BSC methodology has appeared regularly since the beginning of the millennium, where its use was from 48% (2010) to 34% (2017) in industrial enterprises in developed countries. In the Slovak environment, authors [51] conducted an analysis of using the Balanced Scorecard according to the individual sectors of the economy. It results from their research in 2017 that the BSC is used by 8.45% of industrial enterprises in Slovakia.

These studies are described in more detail in literature review and discussion.

## 3. Materials and Methods

The main objective of this research is to verify the significance of applying non-financial factors when managing enterprises in the engineering industry and to verify the dependence between using Balanced Scorecard and the lack of human and financial resources for its usage in the given enterprises. The goal was determined related to the defined hypotheses that were subsequently verified with statistical methods. To obtain data for our research, the method of a questionnaire that we consider to be the most suitable tool for this type of research was chosen. The questionnaire is an optimal data source for our research since it is compiled according to the content, logical, and psychological set of questions through which it is possible to obtain opinions of selected respondents on issues that are the research subjects. The research was conducted in an online form through a google form.

The construction of questions in the questionnaire was created with the possibility of choosing one or more answers to each question and there were also questions asked in the form of a Likert scale. By the questions asked in the Likert form, the answers were given points by the scale of five options when respondents could choose whether they agree or disagree with a given statement. The Likert scale also had a possibility of choice for respondents to indicate the possibility of not being able to comment.

The questionnaire was comprised of two sections—the identification and the research section. The first part included identification data about companies, i.e., the size of an organization, the enterprise ownership, its territorial area of activity. The second section was focused on the information about enterprise strategy, the issue of management and measurement of enterprises' performance and the questions related to the usage of non-financial indicators with a focus on the Balanced Scorecard methodology (see Appendix A).

The questionnaire research was carried out from January to June 2019. The questionnaire was dispatched to companies operating in the engineering sector that are classified according to the OKEČ classification—Sectoral classification of economic activities and SK NACE—Statistical Classification of Economic Activities in the European Community, SECTION C Industrial production [52] classification of enterprises. To address the respondents, the database of Entrepreneur Index [53] that includes the database of companies in Slovakia was selected. The condition for companies being selected for this research was that a company is still active in the current period. When selecting companies we followed the recommendation of the European Commission no. 2003/361/EC, which governs the size of the enterprise in terms of staff and annual budget, the size of these enterprises was in the categories of micro, small, medium, and large enterprises. Ownership of the company was also an important factor in the selection. We made sure that companies with foreign owners, mixed ownerships, and Slovak owners were also addressed. The questionnaire was sent to business owners or top managers.

The respondents were selected through the probability theory which allows finding some random events that are somehow related to some other random events according to the probabilities of these other random events. The probability theory allows us to generalize the results obtained to the whole

sample selected. The results obtained were further analyzed, thus conducting a census. To select the respondents, the method of a random selection of companies where each company has an equal probability of being selected was applied. The companies of random selection must be defined so that every enterprise has the same opportunity to get in the sample selected. As we mentioned, the selection of enterprises in the engineering sector was carried out through the data that are used by the Entrepreneur Index and these data are obtained from freely available databases and registers.

643 companies operating in the field of engineering were addressed from the mentioned database. Out of the respondents addressed, 182 answers used for our research returned back. Thus, the questionnaire return rate was at the level of 28.30%. We consider such a return rate within the questionnaire research as relatively standard and we can consider the results to be relevant.

The data from the respondents necessary for our research were processed through research methods such as descriptive statistics, contingency table, and others, using an analysis, comparison, synthesis, selection, induction, and deduction. The established hypotheses were verified using Pearson's Chi-square Test of Independence. Subsequently, the Chi-square calculation was contrasted with critical values for the error probability selected and the identified level of freedom. The hypothesis was verified using the Chi-square test in statistical programme Statistica from the software company StatSoft version 12.0. The formulas used are shown in Table 1.

**Table 1.** Used formulas with explanatory note.

| Indicator | Formula | Explanatory Note |
|---|---|---|
| **Pearson's Chi-square Test of Independence** | $\chi^2 = \sum \frac{(f_e\ f_t)^2}{f_t}$ | $\chi^2$—the Chi-square value subsequently compared to a table value based on the selected error probability,<br>$f_e$—the empirical frequency of observed variables,<br>$f_t$—the theoretical frequency of observed variables. |
| **Method of proportion of a given phenomenon in population** | $p = \hat{p} \pm z_\alpha * \sqrt{\frac{\hat{p}*\hat{q}}{n}}$ | $\hat{p}$—method of proportion of given phenomenon in the population,<br>$q$—proportion of the opposite phenomenon in the selected sample,<br>$n$—size of sample,<br>$z\_\alpha$—confidence level. |
| **Standard deviation** | $\partial = \frac{\sqrt{(x1-x)2+(x2-x)}+...+(xn-x)2}{n}$ | $\partial$—standard deviation,<br>$X$—average term value,<br>$x_1$—the value of first item,<br>$x_2$—the value of second item,<br>$x_n$—the value of last item,<br>$n$—number of items. |

## 4. Results

The research dealt with the issue of non-financial indicators and their usage in engineering companies. When examining non-financial indicators, the focus was put on the Balanced Scorecard methodology since this methodology is characterized by using non-financial indicators related to the achievement of financial goals that represent the financial indicators. In this context, a hypothesis was defined verifying the dependences between the significance of non-financial indicators and the application of the Balanced Scorecard concept. Based on the data obtained, the given hypothesis was evaluated using the Chi-square Test of Independence. Table 2 shows the findings obtained in the test.

**Table 2.** The results of testing hypothesis.

| Pearson's Chi-Square Test of Independence | | | |
|---|---|---|---|
| **Calculated Value** | **Error Probability** | **Degree of Freedom** | **Critical Value** |
| $p = 0.0422$ | $\alpha = 5\%$ (0.05) | DF = 1.00 | $x^2 = 0.01$ |

The hypothesis focused on the dependences between the importance of non-financial indicators and the Balanced Scorecard usage is shown with the values obtained from the respondents. Figure 1 presents the dependence between the importance of non-financial indicators and the use of the Balanced Scorecard methodology in engineering enterprises confirming the existence of the dependence between individual factors.

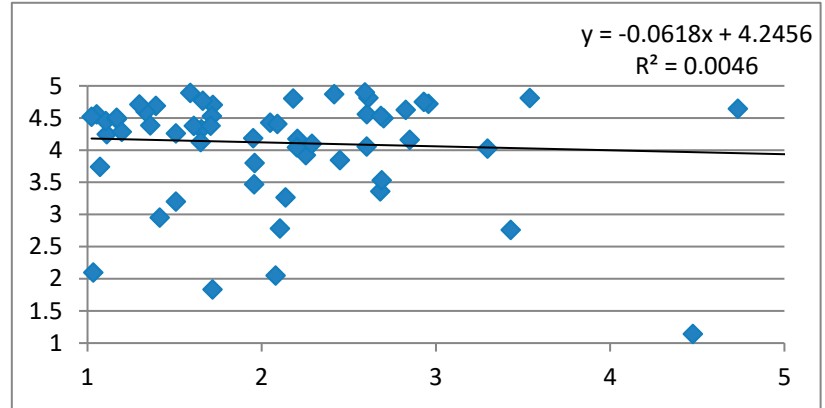

**Figure 1.** Dependences between non-financial indicators and the usage of the Balanced Scorecard (BSC) methodology.

The value calculated from the Chi-square Test of Independence had a lower value than 0.05 which meant that there was a statistically significant relationship between the importance of non-financial indicators and the use of BSC. The given hypothesis was accepted.

The first hypothesis focused on the importance of non-financial indicators related to the usage of BSC. Out of the research studies mentioned in the theoretical part, we found out that using the mentioned methodology in Slovakia is at a relatively low level. We were thus interested in identifying the factors that affect this fact and defining the other hypothesis. The research also focused on the assumption that a statistically significant connection exists between the lack of human resources and the use of BSC in engineering companies. Once again, the Chi-square Test of Independence was used for this calculation. Table 3 presents the resulting test values.

**Table 3.** The results of testing hypothesis.

| Pearson's Chi-Square Test of Independence | | | |
|---|---|---|---|
| **Calculated Value** | **Error Probability** | **Degree of Freedom** | **Critical Value** |
| $p = 0.0446$ | $\alpha = 5\%$ (0.05) | DF = 1.00 | $x^2 = 0.01$ |

Dependences between the lack of human resources and the use of BSC is shown in the graph with the values obtained from the respondents. Figure 2 presents the dependences between the shortage of human resources and the usage of BSC methodology confirming the existence of the dependence between individual factors.

As in a previous case, also in verifying this hypothesis, the assumption was proved when the calculated value was lower than 0.05 proving that the relation of the significance of non-financial indicators and the usage of the Balanced Scorecard methodology is statistically significant. The given hypothesis was accepted in this case.

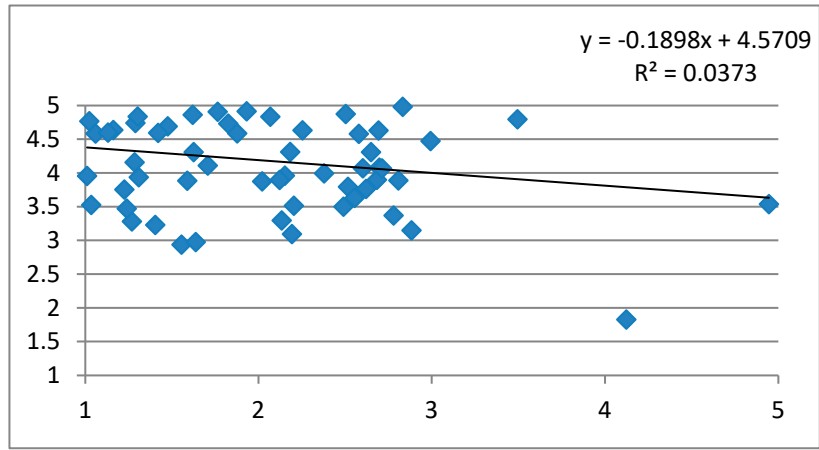

**Figure 2.** Dependences between the shortage of human resources and the use of BSC methodology.

The lack of financial resources thus represents a barrier limiting the usage of the BSC methodology. The research was also interested in whether, apart from financial resources, there is another factor that is responsible for low use of the BSC methodology. Thus, another hypothesis was defined verifying the dependences between the lack of financial resources and the usage of the Balanced Scorecard concept in engineering businesses. Chi-square Test of Independence was used for calculations. The results are given in Table 4.

**Table 4.** The results of testing hypothesis.

| Pearson's Chi-Square Test of Independence | | | |
| --- | --- | --- | --- |
| **Calculated Value** | **Error Probability** | **Degree of Freedom** | **Critical Value** |
| $p = 0.0377$ | $\alpha = 5\% \ (0.05)$ | $DF = 1.00$ | $x^2 = 0.01$ |

The given dependences are shown in Figure 3 by using a scatter plot. This figure also confirms a statistically significant dependence between the lack of financial resources and the use of the BSC methodology in engineering enterprises.

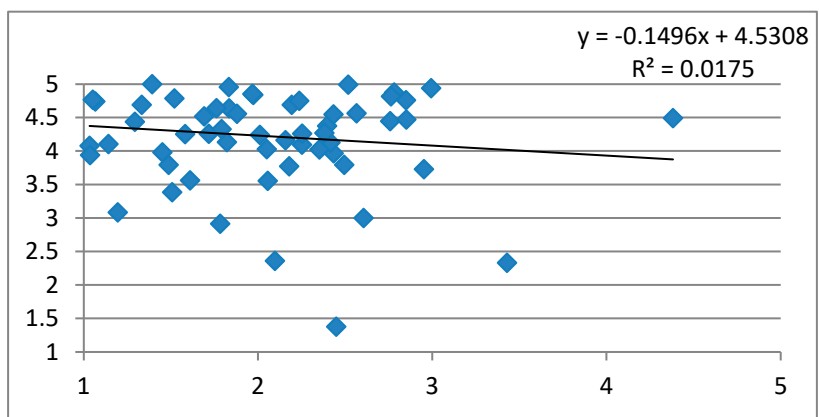

**Figure 3.** Dependences between the lack of financial resources and the use of the BSC methodology.

The final value calculated by the test was lower than 0.05. Based on this value, we can confirm the existence of statistically significant relation of the lack of financial resources and the use of the Balanced Scorecard methodology in engineering enterprises. The hypothesis was thus accepted.

The significant role of using non-financial factors in the management and measurement of enterprise performance is an undeniable fact nowadays. The usage of non-financial indicators in

our conditions is still at a relatively low level. In connection with it, the research was verifying the barriers that hinder the usage of these indicators related to the Balanced Scorecard that is characteristic for using these indicators. The respondents of the research indicated that they consider the lack of non-financial and human resources to be the most significant factors that hinder to use this concept. This fact was also confirmed by verifying hypotheses that were set for this issue.

However, there are also other factors influencing the low usage of the mentioned conception including discrepancies among top managers. Managers have their ideas about managing companies and they are not always willing to make a compromise. A manager must be persuaded also about the support of other managers and lead employees to work in a team spirit. Expressing respect, esteem, encouragement, self-assessment is characterized by the form of evaluation support. From among the least indicated respondents' answers, we can mention the inability to introduce the methodology into the management and the lack of interest in its introduction as barriers of using the Balanced Scorecard methodology.

Figure 4 shows the reasons for not using the Balanced Scorecard methodology compared to the calculation of standard deviation. We can see in Figure 5 that the standard deviation is lowest by the lack of human and financial resources and at the same time, it has the highest values from the respondents.

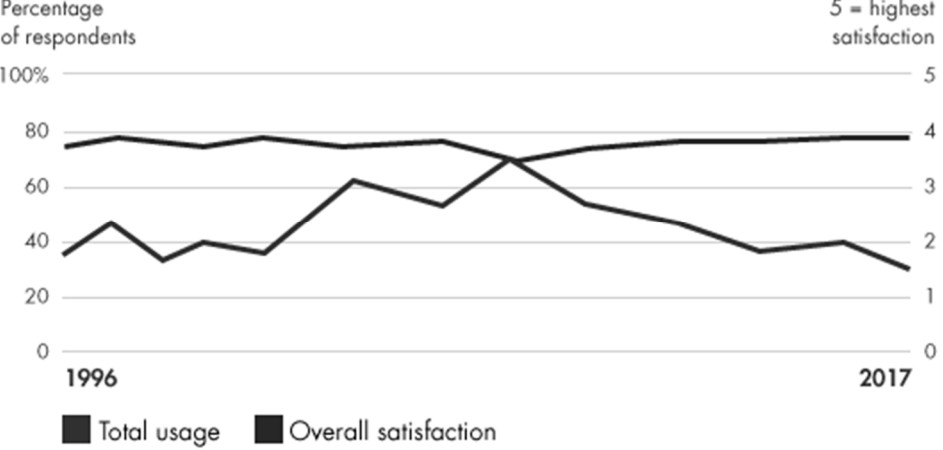

**Figure 4.** Using the BSC methodology and the respondents' satisfaction.

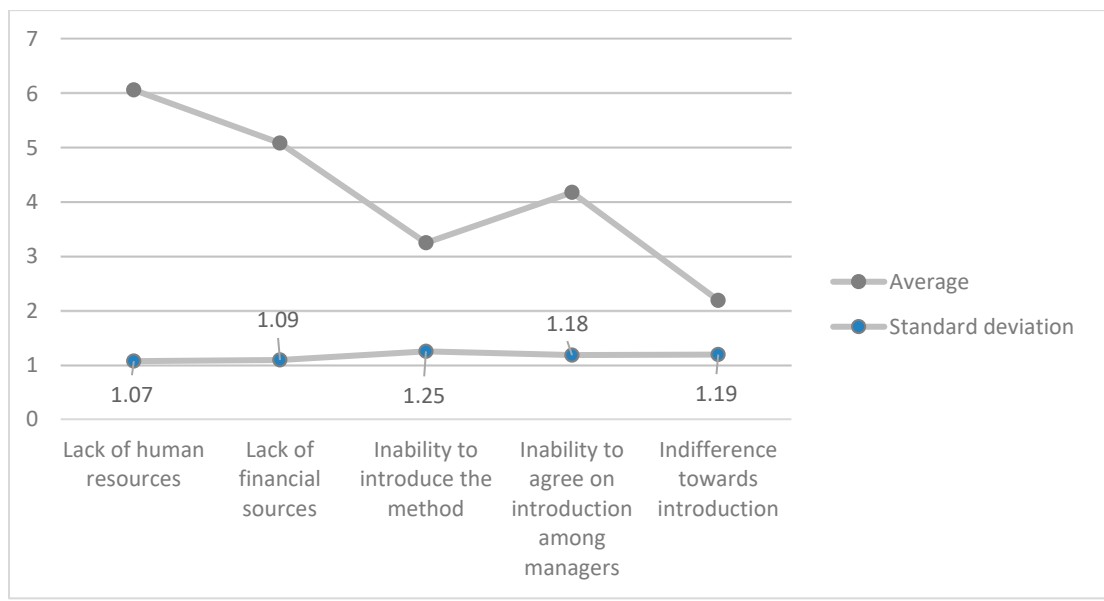

**Figure 5.** Comparison of average and standard deviation.

In this research, we were unable to obtain data from all respondents. In order to consider the results as relevant, we had to use an appropriate method to confirm this relevance. We chose a method of proportioning the phenomenon in the population, which is used when we do not have data from all respondents.

Our research was focused on the BSC in engineering enterprises with the assumption that the concept of Balanced Scorecard is used by more than 20% of engineering companies. The given hypothesis was verified with the methodology of the proportion of a given phenomenon in the population calculated from three given variables. The first variable is the percentage of engineering enterprises that use the Balanced Scorecard concept. The second variable is the percentage of enterprises that do not use this concept and the third variable is represented by the overall number of respondents. The coefficient of the test confidence level was determined by the value of 1,96. The test results are given in Table 5.

**Table 5.** Used formulas with results.

| Indicator | Formula | Explanatory Note |
|---|---|---|
| Method of proportion of a given phenomenon in population | $\hat{p} = 0.2417$ $\hat{q} = 0.7582$ | $p = 0.2417 \pm 1.96 * \sqrt{\frac{0.2417*0.7582}{182}}$ $p = 0.2417 \mp 0.031736$ $0.1795 \leq p \leq 0.3039$ |

The resulting values obtained through the method of the proportion of a given phenomenon range from 17.95% to 30.39%. The values do not reach a minimum level of more than 20% and thus, based on this fact we conclude that the use of the Balanced Scorecard concept does not even reach the anticipated level and the hypothesis was rejected.

## 5. Discussion

The research focused on the issue of non-financial indicators related to the usage of the Balanced Scorecard methodology. This issue is a subject of the research of many experts in the world as well as in the conditions of Slovakia or the Czech Republic. Abroad, regular research is done by a consultancy company Bain and Company which conducted a research in 2018 stating that BSC is used by 53% of companies. In addition, according to Gartner Group, the mentioned conception is used by approximately half of enterprises. A study carried out by the authors [50] presents the usage of BSC by 48% of large and medium enterprises. In the conditions of the Slovak Republic, using the BSC methodology was examined in the author's research [51] who found out that 13.73% of respondents know perfectly this methodology and out of this number, only 9.15% use BSC. In her dissertation thesis, [54] paid attention to the issue of BSC where she made research about learning and using the BSC methodology in Slovakia. According to her research conducted at that time, the methodology of Balanced Scorecard is used only by 6% of enterprises and 12% of enterprises think about its introduction. The study [20] aimed at the use of BSC in small and medium enterprises showed a small usage of the methodology not even reaching 20%.

We can claim that using non-financial indicators when managing a company is still at a relatively low level. [55] found out a statistically positive dependence between strategic management, including also BSC as a strategic managerial tool, and the performance when using non-financial indicators. [56] found a positive correlation between strategy and the use of non-financial performance indicators and their positive impact on business performance. [57] found that businesses using non-financial indicators usually have better performance. Our research found out that the significance of non-financial indicators relates also to the usage of the BSC methodology and the enterprises in the engineering sector that use it consider these indicators to be important. This fact was also confirmed by verifying a hypothesis where a final value $p = 0.0422$ has a lower value than 0.05 confirming the existence of a statistically significant relation between the importance of non-financial indicators and the usage of the

Balanced Scorecard concept. In addition, in verifying the hypothesis focused on the influence of the lack of financial and human resources related to the usage of BSC methodology, the value $p = 0.0446$ and $p = 0.0377$ respectively, thus lower than 0.05. With these values, we confirm that these factors have an effect on using the BSC methodology and there is a statistical dependence between these factors. In regard to using BSC, [46] conducted a research focused on finding out the respondents' satisfaction with the introduction of the Balanced Scorecard methodology. Figure 4 shows the proportion between using the methodology and the satisfaction of companies.

By evaluating the research results, we can state that the companies in the engineering industry that have introduced the Balanced Scorecard methodology consider non-financial indicators in their management being important. The main barriers due to which companies do not use the mentioned methodology include the lack of human and financial resources. These statements were also verified using statistical methods and the given hypotheses were confirmed.

## 6. Conclusions

The research conducted on the factors influencing the use of BSC methodology has yielded partial results and conclusions. The results of the research can serve as a basis for further analysis and definition of factors influencing the use of BSC methodology not only in the industrial sector but also in other sectors. Based on our findings, we know the real usage of the Balanced Scorecard methodology in industrial companies. The obtained data show relatively little use of the BSC methodology, which prompted us to gain further knowledge, namely identifying the barriers that are responsible for this situation. From the sources cited in the previous chapters, we have identified human and financial resources as the main barriers, which was also confirmed by the research. We can conclude that the main objective has been met since, on the basis of statistical verification, the results confirmed the importance of non-financial indicators in the companies surveyed and confirmed the dependence between financial and human resources and the use of BSC methodology. Investigated factors may influence the implementation of BSC methodology serving to improve business performance in both positive and negative meaning.

The research brings results applicable to the users of BSC methodology. In practical terms, factors such as the size of the business and the ownership of the business should be taken into account. Business size is one of the basic assumptions for implementing BSC methodology. Medium and large enterprises are better suited to implement this methodology as BSC contains a number of activities and factors that are not performed to the same extent in micro and small enterprises. These companies have more resources and opportunities to implement, are better prepared for change, have more financial and human resources, and are able to use the experience of previous business performance improvement activities.

From the theoretical point of view, the research brings the possibility to decide on the use of BSC methodology. The study identifies the main factors that influence its use. The research can also serve as a basis for further studies in other sectors or with other factors. The theoretical contribution may also be in defining other factors that may influence the implementation of BSC methodology, including:

- inconsistencies between managers' ideas on the strategic objectives of the company
- managers can only consider BSC methodology as a current trend that will cease to be used over time;
- lack of strategy, which can also be implemented by BSC methodology;
- inability to use the BSC methodology correctly;
- lack of interest in introducing a methodology aimed at improving business performance.

The Balanced Scorecard methodology used was chosen due to non-financial indicators that are the factors contributing to the greater competitiveness of companies in a current turbulent environment. Nowadays, it is no longer appropriate if companies focus only on financial indicators, but it is the strength of employees, innovation and customers' satisfaction that are the main factors of their success.

All these factors involve individual perspectives of the Balanced Scorecard methodology that we consider to be an effective tool for managing a company. The evolution of Balanced Scorecard over the past years has assisted to define principles and factors that were significant from the point of view of users of the organizational procedure [58–60]. The Balanced Scorecard methodology is a managerial tool where all employees, from workers in production to the company's management, can find their strategic aim and can meet it with the right strategic activities and measures [61].

As a result of this research, the following recommendations that could help to increase the interest in the Balanced Scorecard concept and thus increase the interest in management tools in general are presented:

- Conducting studies focused on the issue of managerial tools in regular periods;
- Presenting the studies focused on managerial tools in professional journals and the
- entrepreneurial environment;
- Creating an up-to-date database of enterprises using management tools;
- Building co-operation between enterprises and external and consulting companies focused on the issue of using managerial tools;
- Creating a database of managers who represent a qualified labor force in the issue of management tools.

We can apply the research results in the conditions of the industrial sector in Slovakia. In our opinion, the application of research results is possible mainly in neighboring countries in the region, as the countries operate under similar conditions and have undergone similar developments in the economy.

From the perspective of future research orientation, we would like to focus on modern management tools that meet the concept of sustainable development. Currently, these instruments need to integrate environmental, social, and economic sustainability factors [62]. An important factor to consider is the size and ownership of the business, which influence the use of the methodology used to improve business performance. Another limiting factor is the size of the sample and the choice of sector in which respondents operate. Each sector has its own way of functioning and BSC methodology can meet the requirements for improving business performance in different sectors differently. As another problem we can mention the scope of the study, which was limited to certain variables. In our study, we also outlined other possible issues that could be investigated in the future.

**Author Contributions:** Conceptualization, E.B. and P.G.; methodology, P.G.; software, E.B.; validation, E.B., P.G. and B.B.; formal analysis, J.N.; investigation, B.B.; resources, P.G.; data curation, E.B.; writing—original draft preparation, J.N.; writing—review and editing, B.B.; visualization, J.N.; supervision, P.G.; project administration, E.B.; funding acquisition, B.B. All authors have read and agreed to the published version of the manuscript.

**Funding:** This research was funded by VEGA scientific project No. 1/0134/17 Importance of value orientation—expectations and prospects of the young generation with regard to their employability.

**Conflicts of Interest:** The authors declare no conflict of interest.

## Appendix A

*A Part of the Questionnaire Focused on Balanced Scorecard*

1.  The number of employees in the company:

    (a)  up to 9   (micro-enterprise)
    (b)  10–49   (small enterprise)
    (c)  50–249   (medium enterprise)
    (d)  more than 250   (large enterprise)

2.   According to the ownership, the company is:

(a)   a private company owned by a Slovak owner
(b)   a private company with foreign property participation
(c)   a public company
(d)   mixed (a public company also owned by private owners)

3.   Which sector does your company operate in?

(a)   industry
(b)   commerce
(c)   services
(d)   state administration, i.e., public administration
(e)    other

4.   Does your company measure its performance?

(a)   Yes
(b)   No, the company does not even consider it.
(c)   No, but the company will consider it in the future.

5.   How does your company measure its performance?

(a)   Using modern financial approaches (EVA—Economic value added, MVA—Market value added, CSFROI—Cash flow return on investment, TSR—Total shareholder return, Value Based Management … )
(b)   Total Quality Management—TQM
(c)   Controlling
(d)   Benchmarking
(e)   Model of Excellence
(f)   Balanced Scorecard
(g)   Other way of measuring business performance (please specify)

6.   Please provide a reason why the company is considering measuring its performance:

(a)   Measurement of performance is currently necessary due to market success
(b)   Change in top management thinking
(c)   Plenty of positive examples from the business environment
(d)   Access to new information confirming the importance of measuring business performance (seminars, trainings, lectures … )
(e)   Other reason (please specify)

7.   Do you agree with the statement that, apart from financial indicators, companies should also deal with non-financial indicators?

(a)   Yes, the company considers them important
(b)   No, the company does not consider them important
(c)   I do not know

8.   Do you know the system of performance measurement of Balanced Scorecard?

(a)   Yes, our company uses it
(b)   Yes, our company is considering implementing the Balanced Scorecard system

(c)      Yes, but our company does not use it

(d)      No, but we would be interested in information about the BSC system

(e)      No, our company does not know it

9.    Please, indicate the reasons for not using the Balanced Scorecard methodology in your company:

(a)      The inability to clarify the current strategy since the Balanced Scorecard system is determined by company's strategy

(b)      Insufficient financial resources

(c)      The lack of human resources

(d)      The company considers Balanced Scorecard only a current trend that has no significance for the future

(e)      Inconsistency among business managers' ideas

(f)      Another reason:

10.   Please give reasons why the company is considering the introduction of Balanced Scorecard:

(a)      The importance of implementing a business strategy into practice through Balanced Scorecard

(b)      The company is interested in analyzing its activities not only through financial but also non-financial indicators including Balanced Scorecard

(c)      Balanced Scorecard involves all employees in the process of achieving business goals

(d)      Balanced Scorecard represents a competitive advantage for the company

(e)      Plenty of positive examples from the business environment

(f)      Other reason (please specify)

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
