# Peer review of "Factors Affecting the Use of Balanced Scorecard in Measuring Company Performance"

_sustainability, doi:10.3390/su12031178_

Round 1

Reviewer 1 Report

Review Report: Impact of Non-financial Indicators as an Important Factor in Sustainability business: A Study of Slovak Engineering Sector

Main Comments and Suggestions

I consider that the idea is interesting and worth doing research. The research topic is well explained as concerns its necessity and it fills a gap in the existing literature.

The paper is well organized. However, the paper has some weaknesses that should be improved.

The purpose of the paper should be precisely specified in introduction section as well as main results of the paper. In introduction Authors should stress what is their contribution to the body of knowledge. Remainder of the paper should be added in introduction section.

Literature review is weak. It should be extended. Authors should present what are previous research in the field, what methodology is obtained there, why they choose different methodology, why this methodology is better for this research, what are the results of previous research in the field. Thus, the literature needs to be extended. 

The Authors should also argue why Slovakia and why this specific industry.

The methodology is well explained and appropriately tailored to the research.

The results are presented clearly. The conclusions should underline the paper’s contributions. I suggest emphasizing the originality/value of the study in the abstract.

Thank you for such an interesting research.

Author Response

Dear Reviewer, 

thank you for your valuable comments and professional recommendations for improving the paper. We were trying to do our best to revise it. 

Authors

Reviewer 2 Report

Dear Authors

Important remarks which made me to give you so critic opinion on the paper are as follow. In general, the paper has potential and I like the idea but, in my opinion, needs a lot of work. That is why I would like to encourage the authors to revise their paper following my opinions and resubmit it again. First, English is incomprehensible. So, proofreading by a good native speaker is mandatory. Second, the authors must take care of details not only to write a text but also read all very thoroughly and correct all mistakes – which indeed is a lot in this paper. The authors must read the paper again and revise its sentences and paragraphs as they all must be well developed, described, and logically connected. Third, the authors must tidy their objectives and argumentation, their flow of thoughts, and the way which they are writing about particular problems. The authors must make their research approach and your paper clear for a reader.

Right now it seems like the paper raises more questions than it answers. I am of the view that there are many major issues with this paper that should be addressed ahead of any publication consideration.

First, the title of the paper text is not compatible with the text. First of all, the title is incomprehensible, e.g. what does mean “an impact of …. as….”? – in my view, it must be an impact of sth on sth; what does mean “sustainability business”? – in my view, the author(s) mean sustainable business or sustainability in business, etc. Then, there is about non-financial indicators and sustainability business in Abstract, whereas the text focuses on Balanced Scorecard and its relations with human and financial resources. In addition, the objective of this study indicated in Abstract is not related to the title of the paper and has absolutely nothing to do with sustainability in business and sustainable business. Overall, the authors(s) must rethink the title and objectives of this paper and follow them throughout your paper.  

Second, I believe this topic of sustainable business has a great deal of value for both research and practice, and examination of various issues which can contribute to improvement of such business is needed. Probably, non-financial indicators and Balanced Scorecard can be used to improve sustainable business but it requires argumentation. For example, the Introduction section needs rigorous editing to provide a clearer and tighter set of arguments for the research objective and associated questions. It would be needed to present the problem statement based on the relevant latest literature and with references to the literature. Then the research gap (motivation of the study), the objectives of this study and research questions must be provided (based on this literature review). Then, the research method should be indicated. At the end of Introduction, a structure of the paper should be presented (one short paragraph).

Third, the theory building sections for the study needs greater clarity with respect to writing. Authors indicated three main concepts (constructs) in their study, i.e., non-financial indicators, Balanced Scorecard, and sustainable business; whereas only Balanced Scorecard was only described but sustainable business was not even mentioned and defined. So, these three concepts and relations between them should be presented in-depth based on the recent literature in the Theoretical background section. This presentation must be in the context of the objective and title of the paper, mainly in the context of the importance of non-financial indicators and Balanced Scorecard for sustainable business. It is also necessary to propose the theoretical model comprising the research concepts and hypotheses. Now the hypotheses are not related to sustainability, sustainable business, and an impact of non-financial indicators on sustainable business. Overall, the authors need to do more to provide a coherent argument for the reader with regards their study, its objective, questions, and hypotheses.

Fourth, the author(s) should do a better job of describing the method, unit of analysis, data collection and data analysis. For example, it is not clearly indicated in the paper where measures were sourced. Were these measures adopted from existing studies or were they developed by the authors? If they are sources, references should be provided. If they are newly developed for this study, the author(s) should provide supporting arguments for the measure formulation. In addition, the list of survey questions should be attached to the paper (in Appendix). Then, the discussion how the survey was distributed, how the firms were selected is needed.

Fifth, the research results are related to the research hypotheses, but such hypotheses are not related to sustainability and sustainable business.

Sixth, the Conclusions section requires to be improved. Please give extensive and detailed answers to the questions: What are the contributions of your study? – Explain the contribution of your work more explicit. Describe clearly what are new and unexpected results from your study. In addition, answer the questions: Are the findings unique to some context? Is it universally applicable in any country? What are the implications of the findings for researchers? What are the implications of the findings for practitioners?  What are the limitations of this study? What is the further research?.

Overall, this research study has a deal of potential. However, the paper needs to do more to lay out the motivations for the study and support the theory building.  The paper requires more work to (i) establish the research relevance; (ii) justify the research study (what is the value here for research/practice?) (iii) properly support the theory building with argumentation and appropriate references, and (iv) improve the presentation and readability.

Author Response

Dear Reviewer, 

thank you very much for your expert and valuable suggestions and comments on the paper. The team of authors was trying to do its best to improve it according to your recommendations. 

Best regards, 

Team of authors. 

Reviewer 3 Report

Although I am sympathetic to the research topic proposed in this article, I consider the contribution of this paper is very weak and I cannot find the novelty of this paper.  

I find that the paper not to be of sufficient quality regarding the theoretical positioning and the empirics, and makes an insufficient contribution to the literature in present form.

In general, this paper seems to be less complete. I want the author(s) to take more time to improve the completeness of the paper.

Especially, I have to address serious aspects in this review about the execution of this paper, as illustrated in details below.

The most serious concern is that this study is very similar to a paper titled ‘Non-financial indicators and their importance in small and medium-sized enterprises (Jan Dobrovic, Maya Lambovska, Peter Gallo, Veronika Timkova, 2018)’. The only difference is whether the research targets SMEs or the engineering industry. Especially for example, Figure 1 of this paper is exactly the same as fig. 1 of the paper titled ‘Non-financial indicators and their importance in small and medium-sized enterprises’. Table 1 of this paper is quite similar to fig. 2 of the paper titled ‘Non-financial indicators and their importance in small and medium-sized enterprises’. The hypotheses and methodologies of the two papers are also similar.

Besides, what's worse, the paper didn't even cite it. I'm a little doubtful beyond wondering why the authors didn't refer to the paper.

I suggest that if there are similar papers you'd like to quote, you should cite properly in the text and should reference the quote. Even if the author(s) wants to extend the previous literature, the authors(s) should include the additional information or a new idea in methodology. And such distinctions from previous papers should be specified to indicate what incremental contributions are made.

Other concerns:

Additionally, the paper is not well-written. In fact, the writing throughout the manuscript is generally redundant and not logically connected. The introduction is more an abstract to me. The authors did not make a good summary on why they have incentives to examine such research questions. The motivation for research is not well explained. For example, what insights can you provide based on your finding? Do you have any suggestions to improve the current practice? Adding the above discussion and extend your literature review may help you make more contributions and position your contributions better.

Literature review is patchy and mixes issues. It should be better organized and streamlined. And more academic prior research should be investigated.

As the author(s) points out the limitations of the paper in their conclusions, the author(s) cannot generalize the results about a particular industry to the entire industry.

Additionally, please make the decimal notation consistent.

Author Response

Dear reviewer, 

We value so much your expert knowledge of the issue. We were trying to do everything to revise the paper and to improve the quality of it following your great advice. 

Best regards, 

the team of authors

Reviewer 4 Report

Impact of Non-financial Indicators as an Important Factor in Sustainability Business: A study of Slovak Engineering Sector

Sustainable business is often seen as the efforts of a company to comply with the United Nations sustainability goals and reduce its environmental footprint. The idea is that customers and shareholders appreciate a commitment to sustainability. Also, procurement policies may put emphasis on sustainable products or services, so sustainability may be a tool to secure the competitive advantage, as well as the future of the company.

You could also link this to “Corporate Social Responsibility”. A couple of references, [7] and [15] points in this direction.

Introduction

Line 34, do you really mean “sustainability business” or “sustainable business”?

Line 34, a missing comma after “construct”

In lines 90 and 91, you use “Balanced Scorecard methodology”, in line 93 “Balanced Scorecard method”. Methodology or method? You continue to use "method" throughout the manuscript, but I guess methodology is more appropriate.

Literature review

You do not mention the origin of Balanced Scorecard:

https://www.amazon.com/Balanced-Scorecard-Translating-Strategy-Action/dp/0875846513/ref=sr_1_1?keywords=balanced+scorecard&qid=1576626651&sr=8-1

or

Robert S. Kaplan, David P. Norton: The Balanced Scorecard—Measures that Drive Performance, Harvard Business Review, Jan-Feb, 1992.

You write: “Many problems have been highlighted in the literature regarding the structure of the tool in its implementation process”. You should provide references for each problem, not just two references at the end of the paragraph [33,34].

Table 1: Is that your own findings, or do you have a reference?

I am not sure it is a good idea to put research hypothesis at the end of the literature review. At least, you should use a subheading.

Materials and methods

Line 178 “google” -> “Google”

Line 193-203. References ??

You should elaborate on the response rate. It may have an impact on the results. Respondents having some positive results (using BSC) may be more inclined to answer a survey than possible respondents that are not using BSC. This is a common mistake. A company may not answer because they do not use BSC and feel embarrassed about this when invited to answer the survey. You should discuss the limitations of the low response rate.

Results

Again, this is the problem with a low number of respondents. If the rest is not using BSC, the 20% finding may be down to 20% out of 28% = 5,6% (worst case).

Discussion

The problem is the low response rate.

Conclusions

The low response rate is discussed. The limitations should be elaborated. The results are interesting, but you need to take into account the problem discussed above.

The conclusions seem valid, in spite of the aforementioned objections.

References

[23]-[27] are not

Author Response

Dear reviewer, 

we thank you very much for your advice and great recommendation on how to improve the quality of the paper. We have revised it according to your great suggestions. 

Best regards

the team of authors

Reviewer 5 Report

Article is relevant

Author Response

Dear reviewer, 

thank you very much for your review. 

Best regards, 

the team of athors

Round 2

Reviewer 1 Report

I am happy with the changes made by the authors. I accept the manuscript in present form.

Author Response

Dear reviewer, 

thank you very much. 

Reviewer 2 Report

Dear Authors

I am not impressed by the amount of work done by you in the manuscript enhancement and addressing my concerns. The revised version of the manuscript still needs major improvements according to my comments stressed in my first report.

The title of the paper does not address the text of the paper. For example, (1) BSC concept is the main concept of the text but there is nothing about this concept in the title; (2) “sustainability in business” is in the title but there is nothing about this concept in the text, etc.   

Introduction is too long and out of focus. It should contained just concise overview of the topic, clearly pointed out a research gap (motivation of the study), research goal, research questions, indication of the research method and introduction to the paper structure.

Literature review does not explain main concept of this paper and relations between them.

In my opinion, Materials and Methodology section does not well describe the method, unit of analysis, data collection and data analysis. Additionally, the survey  questionnaire is very simply and it does not be the basis for good research.

Research results cannot be attractive for academics and practitioners  when the literature background and the methodology are not good.

Conclusions section requires to be improved and adjust to the literature and research findings. .

Overall, this research study has a deal of potential. However, the paper needs to do more to lay out the motivations for the study and support the theory building.  The paper requires more work to (i) establish the research relevance; (ii) justify the research study (what is the value here for research/practice?) (iii) properly support the theory building with argumentation and appropriate references, and (iv) improve the presentation and readability.

Author Response

Dear reviewer, 

we are very sorry for not having you impressed by what we have already revised. Well, we were really trying to do our best at that time and in the given conditions. 

We appreciate your comments very much. We have learned many issues connected with writing a good scientific paper. 

We would be very pleased if you would be more impressed by our work this time. 

Once again, thank you very much for your advice. 

The team of authors

Reviewer 3 Report

First of all, I would like to thank the author(s) for all the effort they have put so far in the improvement of the paper.

However, it would have been better to provide a separate answer for each comment and to specify which parts were modified and how, when replying the reviewer's comments.

Additionally, the following issues still require some amendments:

I still wonder if the information in table 1 is necessary for the development of the logic of the study.

More importantly, it seems that the basis for the hypothesis development is still vague. I suggest the author(s) to complement the logic for each hypothesis to clarify how the authors set the hypothesis.

Please check figure number.

Author Response

Dear reviewer, 

thank you very much for your valuable advice and comments. It has helped us to improve the paper very much.  Hopefully, you would be more satisfied this time. We apologize for not seeing the mistakes in the previous text. But we were trying to do our best at that time and in the given conditions. 

All the best

The team of authors

Round 3

Reviewer 2 Report

The paper is better but I am still not impressed by the amount of work done by you in the manuscript enhancement and addressing my concerns. The revised version of the manuscript still needs major improvements according to my comments stressed in my first and second report.

The title of the paper does not address the text of the paper. The main parts of paper are not coherent. The title, objectives and hypothes are not related.

For example, the title is too long and not clear. Is the paper about “dependences between financial and non-financial factors”?  The first sentence of Abstract also stresses “the dependences between financial, non-financial and personal factors”. In Introduction is written: This  paper  aims  to  point  out  the  connection  between  financial  indicators  and  the  Balanced  Scorecard  methodology  and  to verify  the  factors  related to  the  barriers  hindering  the  use  of  this methodology.” etc. What about is your paper?  It is difficult to know.  In my view the paper is about factors influencing BSC and the whole paper should strictly focus on this issue.

All my previous comments should be read carefully again and address in your paper.

Author Response

Thank you very much. 
